# Assembly and comparative analysis of the complete mitochondrial genome of three *Macadamia* species (*M. integrifolia, M. ternifolia* and *M. tetraphylla*)

Yingfeng Niu[1☯], Yongjie Lu[2☯], Weicai Song[2☯], Xiyong He[1], Ziyan Liu[1], Cheng Zheng[1], Shuo Wang[2], Chao Shi[2]*, Jin Liu[1]*

**1** Yunnan Institute of Tropical Crops, Xishuangbanna, China, **2** Qingdao University of Science & Technology, Qingdao, China

☯ These authors contributed equally to this work.
* chsh1111@aliyun.com (CS); liujin2416@163.com (JL)

## Abstract

### Background

*Macadamia* is a true dicotyledonous plant that thrives in a mild, humid, low wind environment. It is cultivated and traded internationally due to its high-quality nuts thus, has significant development prospects and scientific research value. However, information on the genetic resources of *Macadamia* spp. remains scanty.

### Results

The mitochondria (mt) genomes of three economically important *Macadamia* species, *Macadamia integrifolia*, *M. ternifolia* and *M. tetraphylla*, were assembled through the Illumina sequencing platform. The results showed that each species has 71 genes, including 42 protein-coding genes, 26 tRNAs, and 3 rRNAs. Repeated sequence analysis, RNA editing site prediction, and analysis of genes migrating from chloroplast (cp) to mt were performed in the mt genomes of the three *Macadamia* species. Phylogenetic analysis based on the mt genome of the three *Macadamia* species and 35 other species was conducted to reveal the evolution and taxonomic status of *Macadamia*. Furthermore, the characteristics of the plant mt genome, including genome size and GC content, were studied through comparison with 36 other plant species. The final non-synonymous (Ka) and synonymous (Ks) substitution analysis showed that most of the protein-coding genes in the mt genome underwent negative selections, indicating their importance in the mt genome.

### Conclusion

The findings of this study provide a better understanding of the *Macadamia* genome and will inform future research on the genus.

**Data Availability Statement:** The assembled mitochondrial genome sequence and annotation have been deposited at GenBank (https://www.

ncbi.nlm.nih.gov/genbank/) under the accession MW566570, MW566571 and MW566572.

**Funding:** This work was supported by the National Natural Science Foundation of China (No. 31760215 and No. 31801022) and the Technology Innovation Talents Project of Yunnan Province (2018HB086). The funders had no role in study design, data collection and analysis, decision to publish, or preparation of the manuscript.

**Competing interests:** The authors have declared that no competing interests exist.

## 1. Introduction

*Macadamia* spp belongs in the family Proteaceae, class *Magnoliopsida*, and order *Proteales*. The Proteaceae family has five subfamilies, 80 genera, and over 1600 species [1, 2]. Most of them are distributed in Oceania and South Africa, while a few are produced in East Asia and South America. Notably, more than 100 species in the Proteaceae family produce flowers that are traded internationally [3]. Besides, the species grown in the northeastern part of Oceania are also rich in nuts. The genus *Macadamia* comprises four species: *Macadamia integrifolia*, *M. jansenii*, *M. ternifolia*, and *M. tetraphylla*. These species are naturally distributed in the subtropical rain forests from southeastern Queensland in Australia to northeastern New South Wales [4, 5]. Among them, *M. integrifolia* and *M. tetraphylla* produce edible nuts; thus, most commercial cultivars are either these two species or their hybrids. The other two species, *M. Jansenii* and *M. ternifolia* produce non-edible nuts containing high levels of bitter cyanide glycosides, thus has not been used to guide the breeding [6, 7]. *Macadamia* seeds are sweet with high nutritional and medicinal value. Therefore, they have enjoyed the reputation of "King of Thousand Fruits". They are also used in international transactions due to their high economic value [8].

Mitochondria (mt) are organelles that primarily convert biomass energy in living cells into chemical energy to fuel biological activities [9]. Additionally, they participate in other biological processes, including cell differentiation, cell apoptosis, cell growth, and cell division [10–13]. Therefore, mt are central to life activities within individual cells and the entire living body [14]. Both plastids and mt harbor genetic information and are thought to have evolved through endosymbiosis of freely living bacteria [15–17]. In most seed plants, nuclear genetic information is inherited from both parents, while cp and mt are derived from maternal genes [18]. Thus, we can temporarily ignore the influence of paternal genes, thereby reducing the difficulty of genetic research and promoting the research of genetic mechanisms [19].

Studies have shown that the size of the mt genome varies significantly between different species. For example, plants have a larger mt genome than animals [20]. Furthermore, mt genome size in seed plants can vary by at least one order of magnitude ranging from ~ 222 bp in *Brassica napus* [21] and ~ 316 Kb in *Allium cepa* [22] to ~ 3.9 Mb in *Amborella trichopoda* [23] and a striking ~ 11.3 Mb in *Silene conica* [24]. This phenomenon may be caused by the abundance of non-coding regions and repeated elements in the plant mt genome [25]. DNA recombination between homologous sequences produces small circular sub-genomic DNA. The circular genomic DNA coexists with the complete "master" genome in the cell. These genomes typically have several kb repeats, leading to multiple heterogeneous forms of the genome [26–31]. The mutation rate of plant mt genomes is very low; however, their rearrangement rate is so high that there is almost no conservation of synteny [32–34].

The development of cost-effective and more efficient DNA sequencing methods like high-throughput sequencing has accelerated mt genome sequencing. So far (until June 2021), the mt genomes of 618 green plant species have been released in the NCBI (https://www.ncbi.nlm.nih.gov/) database. Long-term mutually beneficial symbiosis caused the mt to lose some of the original DNA, possibly by transfer, leaving only the DNA encoding it [35, 36]. Mt DNA integrates DNA from various sources by intracellular and horizontal transfer [37]. Therefore, regardless of the length, gene sequence and content, mt genome varies remarkably among different plant species [33]. The mt genome length of the smallest terrestrial plant is about 66 Kb, and that of the largest terrestrial plant is 11.3 Mb [24, 38, 39]; the number of genes is usually between 32 and 67 [40]. In this study, the mt genomes of three *Macadamia* species were sequenced, assembled, and annotated. Also, their genomic and structural features were analyzed and compared with other angiosperms (and gymnosperms). This study improves our

understanding of *Macadamia* genetics and provides crucial data to inform future research on the evolution of mt genomes of land plants.

## 2. Materials and methods

### 2.1 Genome sequencing

The three *Macadamia* species examined in this study were collected from Yunnan Institute of Tropical Crops (Xishuangbanna, China; 101˚28' E, 21˚92' N). Total genomic DNA was extracted from fresh leaves using modified CTAB [41]. Meanwhile, the quantity and quality of extracted DNA was assessed by spectrophotometry and the integrity was evaluated using a 1% (w/v) agarose gel electrophoresis. The qualified DNA samples were used for Illumian DNA library construction, according to the standard procedure. Subsequently, a paired-end sequencing library with an insert size of 350 bp was constructed. The Illumina Hiseq 4000 high-throughput sequencing platform was used for sequencing. The sequencing strategy involved PE150 (Pair-End 150) and the sequencing data volume of not less than 1 Gb. Illumina high-throughput sequencing results initially existing as original image data files were con-verted into Raw Reads. CASAVA software was used for Base Calling.

### 2.2 Genome assembly and annotation

SPAdes v.3.5.0 [42] software was used to splice and assemble mt genome sequences. To correct the splicing results, the raw sequencing data were mapped to mitochondrial sequences using Geneious software [43]. DOGMA [44] and NCBI were used to annotate the mt genome. The Blastn and Blastp method was used to compare mt gene-encoding protein and rRNAs among related species. TRNA scan-SE2.0 [45] and ARWEN [46] were used to annotate tRNA. The tRNAs with unreasonable length and incomplete structure were eliminated. Subsequently, a tRNA secondary structure diagram was generated. The final mt genomes of *M. integrifolia*, *M. ternifolia*, and *M. tetraphylla* have been deposited in the GenBank (Accession number: MW566570/MW566571/MW566572).

### 2.3 Analysis of repeat structure and sequence

Microsatellites within the mt genomes of the three *Macadamia* species were identified using MISA [47, 48]. The minimum number of repeats for the motif length of 1, 2, 3, 4, 5, and 6 were 10, 6, 5, 4, 3, and 3, respectively, were identified in this analysis. The tandem repeats were detected using Tandem Repeats Finder v4.09 software [49] with default parameters.

### 2.4 DNA transformation from cp to mt and RNA editing analyses

The cp genome of *M. integrifolia* (NC_025288) was downloaded from the NCBI database. Chloroplast-like sequences were identified and the genome was mapped using TBtools [50]. The online program Predictive RNA Editor for Plants (PREP) suite [51] was adopted to iden-tify the possible RNA editing sites in the protein-coding genes of the three *Macadamia* species. The cutoff value was set as 0.2 to ensure accurate prediction. The protein-coding genes from other plant mt genomes were used as references to reveal the RNA editing sites in the mt genomes of the three *Macadamia* species.

### 2.5 Phylogenetic tree construction and Ka/Ks analysis

The genome sequences of the three *Macadamia* species were compared with those of 35 (S1 Table) other plant species to further verify their phylogenetic position. Notably, the complete mt genome sequences of these species were available in the NCBI database. Phylogenetic

analyses were performed on 23 conserved protein-coding genes (*atp1*, *atp4*, *atp6*, *atp8*, *atp9*, *ccmB*, *ccmC*, *ccmFc*, *ccmFn*, *cob*, *cox1*, *cox2*, *cox3*, *matR*, *nad1*, *nad2*, *nad3*, *nad4*, *nad4L*, *nad5*, *nad6*, *nad7* and *nad9*) that were extracted from the mt genomes of the 35 plant species using TBtools [51]. These conserved genes were then aligned using Muscle [52] implemented in MEGA X [53]; the alignment was modified manually to eliminate gaps and missing data. The GTR + G + I model was determined to be the best model based on the Akaike Information Criterion (AIC) and Bayesian Information Criterion (BIC) calculated by ModelFinder [54]. The Maximum Likelihood (ML) algorithm in MEGA X [53] was used to construct a phylogenetic tree. The bootstrap consensus tree was inferred from 1000 replications. *Cycas taitungensis* and *Ginkgo biloba* were designated as the outgroup in this analysis.

The Ka and Ks replacement rates of protein-coding genes in mitochondrial genomes of the three *Macadamia* species and other higher plants were analyzed. blastn in TBtools was used to extract the sequences of corresponding protein-coding genes in *Macadamia* and *N. nucifera* genomes. The Ka and Ks replacement rates of each protein-coding gene were estimated using *N. nucifera* genome as a reference.

## 3. Results and discussion

### 3.1 Genomic features of the mt genomes of the three *Macadamia* species

The mt genomes of *M. integrifolia*, *M. ternifolia* and *M. tetraphylla* have a typical terrestrial plant genome ring structure (Fig 1). A total of 71 unique genes were identified in the mt genomes of the three *Macadamia* species, including 42 protein-coding, 26 tRNA, and 3 rRNA genes (Table 1). In addition, two copies of *rRNA26*, *ccmB*, *rps19*, *trnN*-GTT, and *trnH*-GTG, and seven copies of *trnM*-CAT were identified. It has been established that the mt genomes of land plants contain a variable number of introns [55]. In the present study, the three mt genomes had ten genes with introns, length ranging from 13 bp (*rps3*) to 31,841 bp (*cox2*) where *ccmFC*, *rpl2*, *rps3*, and *rps10* had two introns, *cox2* had three, *nad1*, *nad4*, and *nad5* had four and *nad2* and *nad7* had five introns. Besides, in all protein-coding genes, except *atp6*, *cox1*, *nad1*, *nad4L*, *rps4*, and *rps10*, which had ACG as the start codon, all the others had ATG as their start codon. In addition, the stop codons in all the protein-coding genes were: TAA 45.2%, TGA 28.6%, TAG 14.3%, CAA 9.5%, and CGA 2.4%.

The size and GC content of mt genome are the primary characteristics. Here, we compared the size and GC content of mt genomes between three *Macadamia* species and 36 other green plants, including four phorophytes, three bryophytes, two gymnosperms, four monocots, and 23 dicots (S1 Table). The size of the mt genomes ranged from 22,897 bp (*Chlamydomonas moewusii*) to 2,709,526 bp (*Cucumis melo*) (Fig 2). Compared to phorophytes and bryophytes, the mt genomes of the three *Macadamia* species are larger. The GC content in the mt genomes was also highly variable, ranging from 32.24% in *Sphagnum palustric* to 50.36% in *Ginkgo biloba*. Overall, the GC content of angiosperm mt genome (including monocots and dicots) is higher than that in bryophytes but less than in gymnosperms [56, 57], implying that the GC contents fluctuated following the angiosperms divergence from bryophytes and gymnosperms. Interestingly, the GC content significantly fluctuated in algae and was mostly conserved in angiosperms, although their genome sizes vary significantly.

### 3.2 Repeat sequences analysis

Microsatellites or simple sequence repetitions (SSRs) are DNA fragments composed of short sequence repeating units of 1–6 base pairs [58]. Their unique value is created by their polymorphism, relative abundance, codominant inheritance, large-scale genome coverage, and PCR detection simplicity [59]. Based on the SSRs analysis, we identified 87 SSRs with SSRs

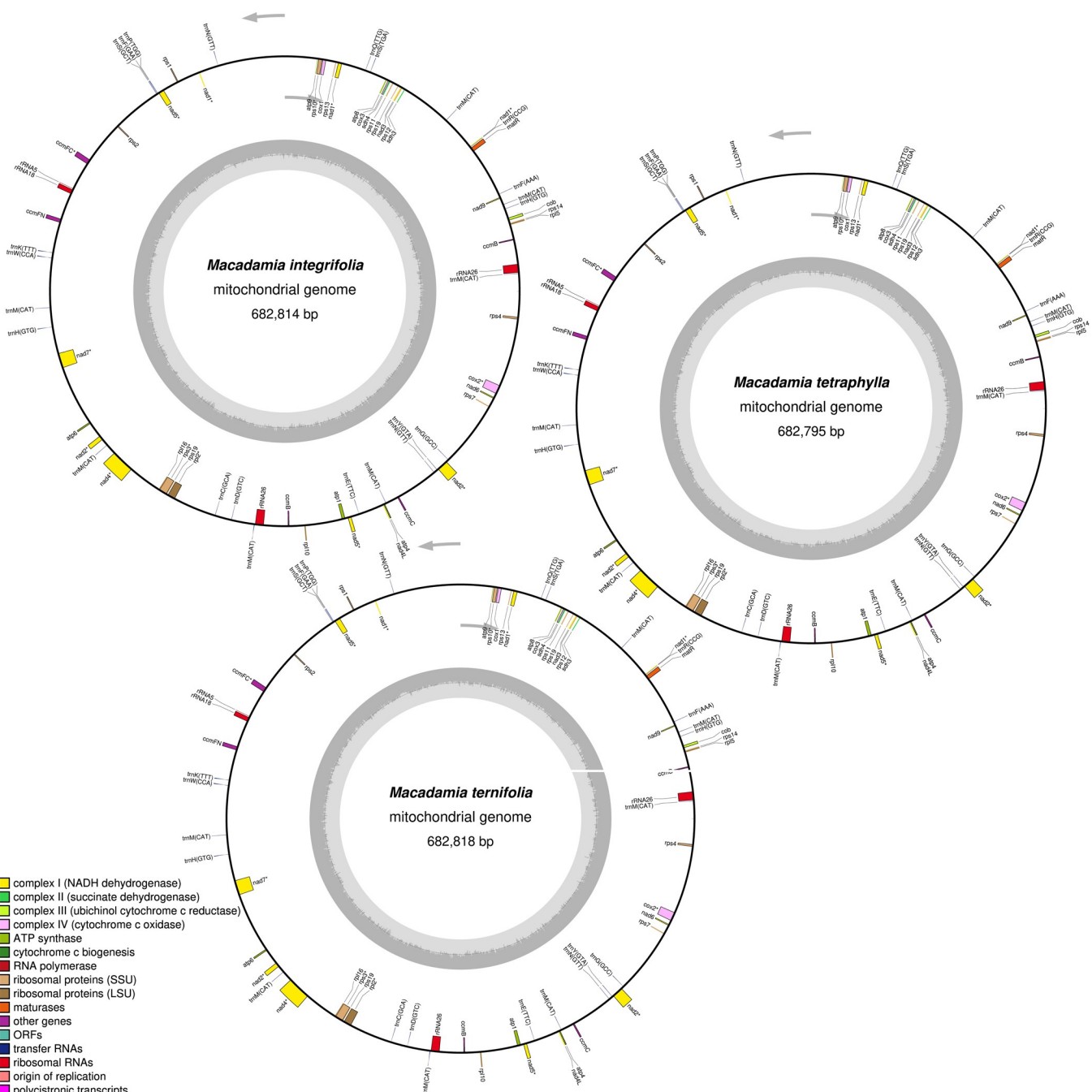

**Fig 1. The circular map of three *Macadamia species* mitochondrial genome.** Gene map showing 71 annotated genes of different functional groups.

monomers and dimers accounting for 70.11% of the total SSRs. Adenine (A) was the most repeated monomer with 19 (38%) out of the 50 identified monomer SSRs. The AT repeat was the most common dimer SSR, accounting for 66.67% of all the identified dimers. However, one hexamer [ATTAGG(X3)] was present in the mt genomes of three *Macadamia* species.

Among the reference mt genome, only *Nelumbo nucifera* has been published in the NCBI database. *N. nucifera* belongs to the family Nelumbonaceae and the same order (Proteales) with *Macadamia*. Therefore, the mt genome of *N. nucifera* was used as a reference for

**Table 1. Gene profile and organization of three *Macadamia* species (*M. integrifolia*, *M. ternifolia* and *M. tetraphylla*).**

| Group of genes | Gene/element | Size(bp) | GC_Percent | AminoAcids (aa) | InferredInitiation Codon | Inferred TerminationCodon |
|---|---|---|---|---|---|---|
| ATP synthase | atp1 | 1530 | 45.29% | 509 | ATG | TGA |
| | atp4 | 597 | 43.05% | 198 | ATG | TAG |
| | atp6 | 783 | 39.08% | 260 | ACG | TAA |
| | atp8 | 480 | 40.63% | 159 | ATG | TAA |
| | atp9 | 225 | 46.67% | 74 | ATG | CAA |
| Cytochrome c biogenesis | ccmB(2) | 621,621 | 42.83% | 206 | ATG | TGA |
| | ccmC | 771 | 44.23% | 256 | ATG | TAA |
| | ccmFCa | 1356 | 46.53% | 451 | ATG | TAA |
| | ccmFN | 1734 | 47.58% | 577 | ATG | TGA |
| Ubichinol cytochrome c reductase | cob | 1182 | 42.39% | 393 | ATG | TGA |
| Cytochrome c oxidase | cox1 | 1584 | 44.26% | 527 | ACG | TAA |
| | cox2a | 822 | 42.34% | 273 | ATG | TAG |
| | cox3 | 798 | 45.11% | 265 | ATG | TGA |
| Maturases | matR | 1968 | 52.64% | 655 | ATG | TAG |
| NADH dehydrogenase | nad1a | 978 | 44.99% | 325 | ACG | TAA |
| | nad2a | 1467 | 40.90% | 488 | ATG | TAA |
| | nad3 | 357 | 41.74% | 118 | ATG | TAA |
| | nad4a | 1488 | 42.67% | 495 | ATG | TGA |
| | nad4L | 303 | 37.29% | 100 | ACG | TAA |
| | nad5a | 1989 | 41.78% | 662 | ATG | TAA |
| | nad6 | 630 | 40.95% | 209 | ATG | TGA |
| | nad7a | 1185 | 45.23% | 394 | ATG | TAG |
| | nad9 | 573 | 42.93% | 190 | ATG | TAA |
| Ribosomal proteins (LSU) | rpl2a | 999 | 52.15% | 332 | ATG | CAA |
| | rpl5 | 561 | 44.74% | 186 | ATG | TAA |
| | rpl10 | 516 | 46.32% | 171 | ATG | TAA |
| | rpl16 | 492 | 43.09% | 163 | ATG | TAA |
| Ribosomal proteins (SSU) | rps1 | 606 | 43.56% | 201 | ATG | TAA |
| | rps2 | 648 | 39.20% | 215 | ATG | CAA |
| | rps3a | 1692 | 43.91% | 563 | ATG | TAG |
| | rps4 | 1059 | 40.51% | 352 | ACG | TAA |
| | rps7 | 447 | 43.18% | 148 | ATG | TAA |
| | rps10a | 333 | 39.04% | 110 | ACG | CGA |
| | rps11 | 444 | 45.27% | 147 | ATG | CAA |
| | rps12 | 378 | 45.50% | 125 | ATG | TGA |
| | rps13 | 351 | 39.60% | 116 | ATG | TGA |
| | rps14 | 303 | 40.92% | 100 | ATG | TAG |
| | rps19(2) | 285,285 | 40.00% | 94 | ATG | TAA |
| Transport membrane protein | sdh3 | 336 | 37.20% | 111 | ATG | TGA |
| | sdh4 | 450 | 41.33% | 149 | ATG | TGA |
| Ribosomal RNAs | rrn5 | 119 | 52.94% | | | |
| | rrn18 | 2061 | 55.12% | | | |
| | rrn26(2) | 3989,3989 | 53.02% | | | |

*(Continued)*

**Table 1.** (Continued)

| Group of genes | Gene/element | Size(bp) | GC_Percent | AminoAcids (aa) | InferredInitiation Codon | Inferred TerminationCodon |
|---|---|---|---|---|---|---|
| *Transfer RNAs* | trnR-CCG | 75 | 57.33% | | | |
| | trnN-GTTb(2) | 75,72 | **49.33%** | | | |
| | trnD-GTCb | 74 | 63.51% | | | |
| | trnC-GCA | 76 | 52.63% | | | |
| | trnQ-TTG | 72 | 47.22% | | | |
| | trnE-TTC | 72 | 50.00% | | | |
| | trnG-GCC | 74 | 54.05% | | | |
| | trnH-GTGb(2) | 75,75 | 54.67% | | | |
| | trnK-TTT | 75 | 46.67% | | | |
| | trnM-CATb(7) | 72,75,73,72,77,72,72 | 59.72%,46.67%,43.84%, 59.72%,44.16%,59.72%,59.72% | | | |
| | trnF-AAA | 75 | 49.33% | | | |
| | trnF-GAA | 74 | 47.30% | | | |
| | trnP-TGG | 75 | 54.67% | | | |
| | trnS-TGA | 88 | 51.14% | | | |
| | trnS-GCT | 91 | 46.15% | | | |
| | trnW-CCAb | 74 | 51.35% | | | |
| | trnY-GTA | 84 | 51.19% | | | |

Notes: The numbers after the gene names indicate the duplication number. Lowercase a indicates the genes containing introns, and lowercase b indicates the chloroplast-derived genes.

comparative analysis in the present study. The monomers in *N. nucifera* were lower than in the three *Macadamia* species, while pentamers and hexamers in *N. nucifera* were significantly higher than in the three *Macadamia* species (Fig 3A). Moreover, the SSRs in mt genomes of *M. integrifolia*, *M. ternifolia*, *M. tetraphylla*, and *N. nucifera* were mainly single-nucleotide A/T motifs, and dimer AT/TA motifs. Within the *Macadamia* genus, the mt SSRs among the

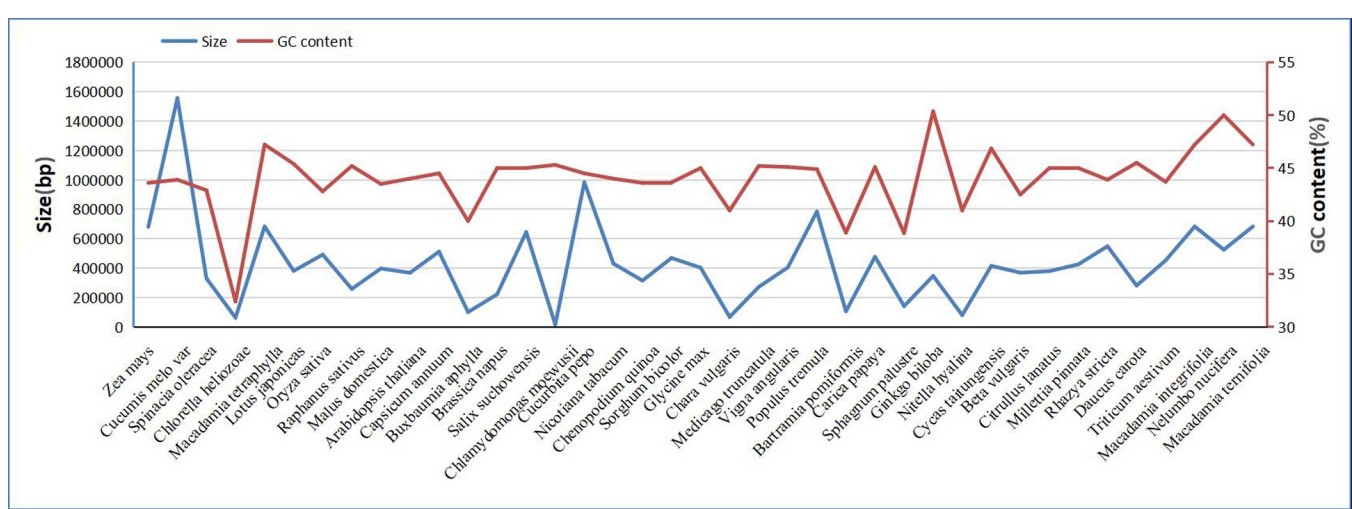

**Fig 2. The sizes and GC contents of 39 plant mitochondrial genomes.** The blue dots represent the genome size and the orange trend line shows the variation of GC content across the different taxa.

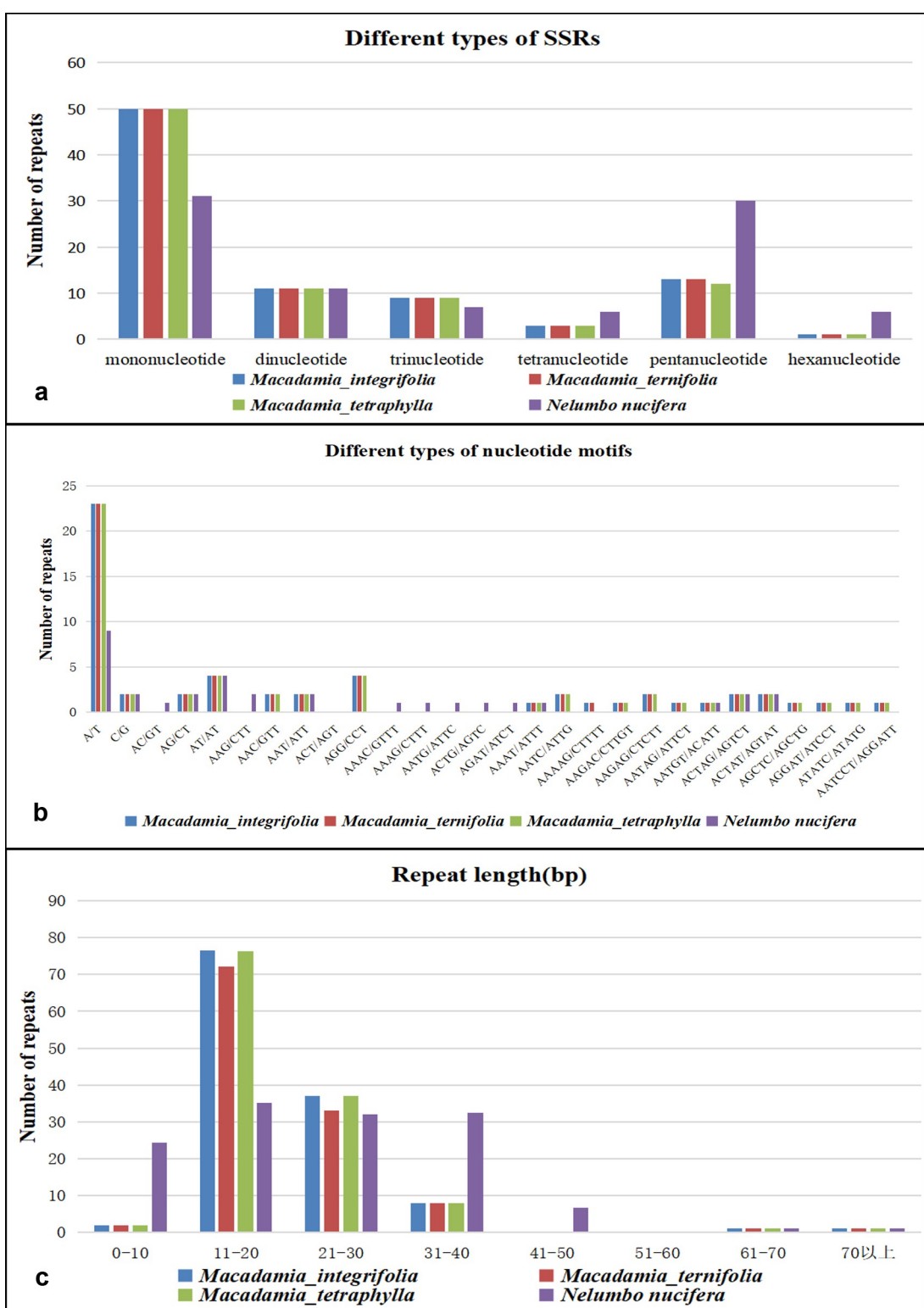

**Fig 3. The comparison of microsatellites and oligonucleotide repeats in three *Macadamia* species and *N. nucifera* mitochondrial genomes.**

different species are highly similar (Fig 3B). However, compared with *N. nucifera*, there were both differences and similarities. For example, the single nucleotide A/T in the three *Macadamia* species has 23-unit repeats, while *N. nucifera* has only nine. Nevertheless, their single-nucleotide C/G numbers were the same (two-unit repeats) (Fig 3B). In addition, the AG/CT and AT/AT motifs unit repetitions are the same, although *N. nucifera* also has an AC/GT motif, lacking in the three *Macadamia* species. Interestingly, the pentanucleotide AATGT/ACATT, ACTAG/AGTCT, and ACATT/AGTAT also had the same number of repetitions in the three *Macadamia* species and *N. nucifera*. Overall, the greater the nucleotide motif, the greater the difference between the three *Macadamia* species and *N. nucifera*.

Core repeating units ranging from 1 to 200 bases (tandem repeats) are widely present in eukaryotes and some prokaryotes genomes [60]. In the present study, 25, 21, and 20 tandem repeats (10–33 bp) were identified in the *M. integrifolia*, *M. ternifolia*, and *M. tetraphylla* with a match greater than 95% (S2–S4 Tables). The tandem repeats (11–20 bp and 21–30 bp) significantly varied among the three *Macadamia* species (Fig 3C), where *M. ternifolia* had the least number of repetitions, while *M. integrifolia* and *M. tetraphylla* had a very similar number of repetitions. However, *N. nucifera* had the least (11–20 bp and 21–30 bp) and had the highest (0–10 bp, 31–40 bp, 41–50 bp) tandem repeated compared to the three *Macadamia* species. Besides, no repetitions ranged from 51–60 bp among the four genomes, while the number of repetitions was the same for 60–70 bp and above.

### 3.3 The prediction of RNA editing

RNA editing is a post-transcriptional process entailing the addition, deletion, or conversion of bases in the coding region of a transcribed RNA. The conversion of cytosine to uridine is common in cp and mt genomes of plants [61–65], which improves protein preservation in plants. The accurate detection of ribonucleic acid editing is inseparable from the proteomics data. In the present study, we predicted 42 protein-coding genes (including two multi-copy genes: *ccmB* and *rps19*) in the mt genomes of the three *Macadamia* species using the PREP-mt program [51]. The findings revealed that the RNA editing sites were 688, 689, and 688 (Fig 4).

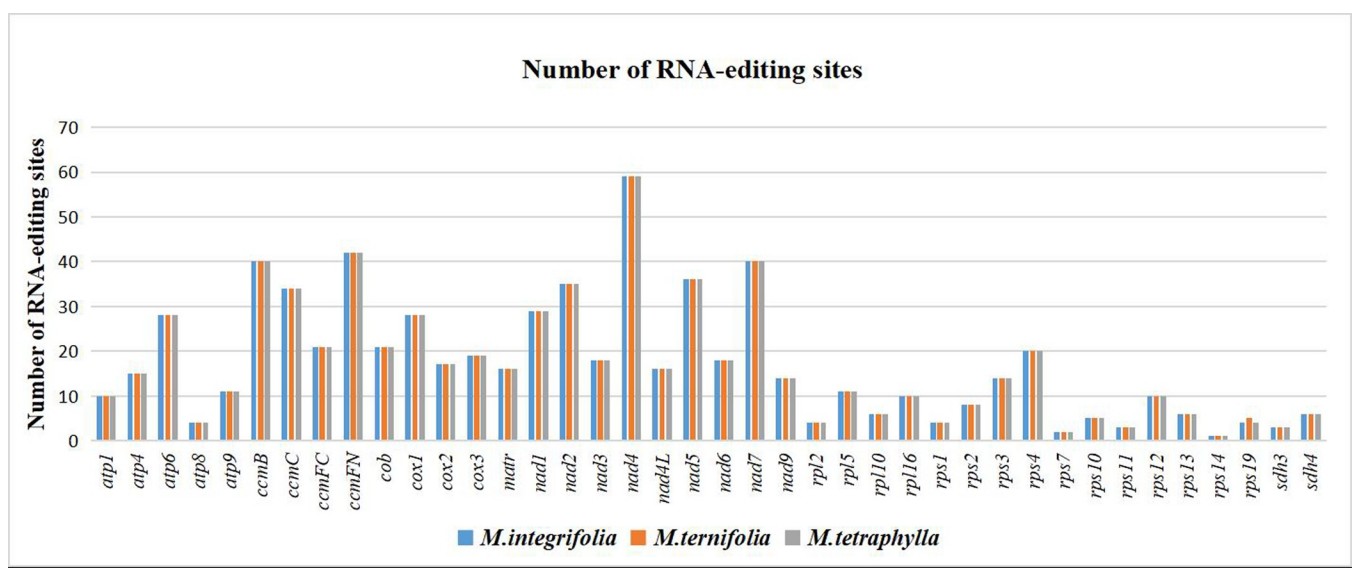

**Fig 4. The distribution of RNA-editing sites in the mt protein-coding genes of three species of *Macadamia*.** The bars of different colors represent the number of RNA-editing sites of each gene.

Among the protein-coding genes, *nad4* had the most RNA editing sites (59 sites), while *atp8*, *rpl2*, *rpl10*, *rps1*, *rps2*, *rps7*, *rps10*, *rps11*, *rps13*, *rps14*, *rps19*, *sdh3*, and *sdh4* had less than 10 RNA editing sites. 236 RNA editing sites occurred in the first base position of the codon, 472 sites appeared in the second base position, and there was no RNA editing in the third base position. *M. ternifolia* had more than one RNA editing site, unlike the other two *Macadamia* species.

The RNA editing increases the diversity at the start and stop codons in protein-coding genes. However, even with RNA editing, 30.2% (208 positions) of amino acid hydrophobicity and 12.5% (86 positions) of amino acid hydrophilicity remained unchanged in the *M. integrifolia* and *M. tetraphylla* mt genomes. However, 6.7% (46 positions) of amino acids were converted from hydrophobic to hydrophilic, and 47.9% (330 positions) from hydrophilic to hydrophobic. In addition, five amino acids were converted from glutamine to stop codons and two from arginine to stop codons (Table 2). The findings in this study revealed that most

**Table 2. Prediction of RNA editing sites.**

| Type | RNA-editing | Number | Percentage |
|---|---|---|---|
| hydrophobic | GCA (A) = > GTA (V) | 1 | 30.23% |
| | GCG (A) = > GTG (V) | 6 | |
| | GCT (A) = > GTT (V) | 4 | |
| | CTC (L) = > TTC (F) | 7 | |
| | CTT (L) = > TTT (F) | 16 | |
| | CCC (P) = > TTC (F) | 6 | |
| | CCT (P) = > TTT (F) | 14 | |
| | CCA (P) = > CTA (L) | 61 | |
| | CCC (P) = > CTC (L) | 14 | |
| | CCG (P) = > CTG (L) | 44 | |
| | CCT (P) = > CTT (L) | 35 | |
| hydrophilic | CAT (H) = > TAT (Y) | 24 | 12.50% |
| | CAC (H) = > TAC (Y) | 11 | |
| | CGC (R) = > TGC (C) | 15 | |
| | CGT (R) = > TGT (C) | 36 | |
| hydrophobic-hydrophilic | CCA (P) = > TCA (S) | 16 | 8.28% |
| | CCC (P) = > TCC (S) | 13 | |
| | CCG (P) = > TCG (S) | 6 | |
| | CCT (P) = > TCT (S) | 22 | |
| hydrophilic-hydrophobic | CGG (R) = > TGG (W) | 43 | 47.97% |
| | TCC (S) = > TTC (F) | 47 | |
| | TCT (S) = > TTT (F) | 58 | |
| | TCA (S) = > TTA (L) | 101 | |
| | TCG (S) = > TTG (L) | 59 | |
| | ACA (T) = > ATA (I) | 7 | |
| | ACC (T) = > ATC (I) | 1 | |
| | ACG (T) = > ATG (M) | 8 | |
| | ACT (T) = > ATT (I) | 6 | |
| hydrophilic-stop | CGA (R) = > TGA (X) | 2 | 1.02% |
| | CAG (Q) = > TAG (X) | 1 | |
| | CAA (Q) = > TAA (X) | 4 | |

Notes: Compared with the other two species of *Macadamia*, *M. ternifolia* had only one more RNA-editing site (CTT (L) = >TTT (F)).

amino acids were converted from serine to leucine (23.3%, 160 sites), proline to leucine (22.4%), and serine to phenylalanine (15.3%). The remaining 269 RNA editing sites included other RNA editing types, such as Ala-Val, His-Tyr, Leu-Phe, Pro-Phe, Pro-Ser, Arg-Cys, Arg-Trp, Thr-Ile, Thr- Met, Gln-X, and Arg-X (X = stop codon). Compared to *M. integrifolia* and *M. tetraphylla*, *M. ternifolia* only had one more RNA-edited site (Leu-Phe).

### 3.4 DNA migration from cp to mt

The cp-like sequences in the mt genome were detected by comparing against the complete cp genome sequence of *M. integrifolia* obtained from the NCBI database (Fig 5). We detected 28 fragments in the mt genome of *M. integrifolia*, ranging in size from 32 bp to 5,210 bp. The cp-like sequence had 36,902 bp, accounting for 5.4% of the mt genome. Five complete annotated tRNA genes were detected, namely *trnH*-GTG, *trnM*-CAT, *trnW*-CCA, *trnD*-GTC, and *trnN*-GTT, with some fragments of *rrn18* genes. The findings also revealed that 28 insertion regions accounted for 23.2% of the cp genome, including seven complete protein-coding genes (*petL*, *petG*, *ndhE*, *rps15*, *rpl23(X2)*, *rpl2*) and eight complete tRNA genes (*trnH*-GUG, *trnD*-GUC, *trnM*-CAU, *trnW*-CCA, *trnP*-UGG, *trnP*-GGG, *trnI*-CAU, *trnN*-GUU). Besides, several protein-coding genes were also identified, including *psbA*, *rpoB*, *psbD*, *psbC*, *ndhC*, *rpl2*, *ycf2(X2)*, *ndhB*, *rps7(X2)*, *ndhD*, *ndhB* and *ycf1*, and some tRNA genes (*trnI*-GAU, *trnA*-UGC, *trnN*-GUU), which migrated from the cp genome into the mt genome. But, most of these genes lost their integrity during the evolution process, and only their partial sequences were found in the mt genome. Furthermore, most cp-like sequences were located in the spacer region of the mt genome. These findings are consistent with previous research, where during evolution, tRNA genes were more conserved than the protein-coding genes and rRNA genes since they play an important role in mt genome [66].

### 3.5 Phylogenetic analysis within higher plant mt genomes

Australia is the origin and center of diversity of the Proteaceae, and this is distributed across remnant landmasses of the southern supercontinent Gondwana [67]. The order Proteales inclusive of Proteaceae, Platanaceae and Nelumbonaceae was established relatively recently, on the basis of molecular data, and morphological synapomorphies for the order are yet to be identified [68, 69]. Phylogenetic analysis was performed to understand the evolution of the three *Macadamia* species compared to 29 dicots, four monocots, and two gymnosperms (out-groups). The phylogenetic tree was constructed based on the comparisons in the data matrix of 23 conserved protein-coding genes (Fig 6). The findings revealed that the phylogenetic tree strongly supports the separation of Proteales from rosids and asterids, the separation of eudicots from monocots and angiosperms from gymnosperms. The evolutionary relationships among all the taxa separated into 20 families (Leguminosae, Cucurbitaceae, Apiaceae, Apocynaceae, Solanaceae, Rosaceae, Caricaceae, Brassicaceae, Salicaceae, Bataceae, Malvaceae, Vitaceae, Lamiaceae, Nelumbonaceae, Proteaceae, Butomaceae, Arecaceae, Poaceae, Cycadaceae, and Ginkgoaceae) were efficiently deduced in the phylogenetic tree (Fig 6). The *Macadamia* chloroplast genome confirms the placement of this family with the morphologically divergent Plantanaceae (plane tree family) and Nelumbonaceae (sacred lotus family) in the basal eudicot order Proteales [70]. In addition, Phylogenetic analysis of chloroplast genomic variation revealed a latitudinal population structure of wild *M. integrifolia* germplasm, suggesting long-term regional isolation of maternal lineages [71]. Overall, evolutionary analyses of organelle genomes suggest that Proteaceae are most closely related to Nelumbonaceae.

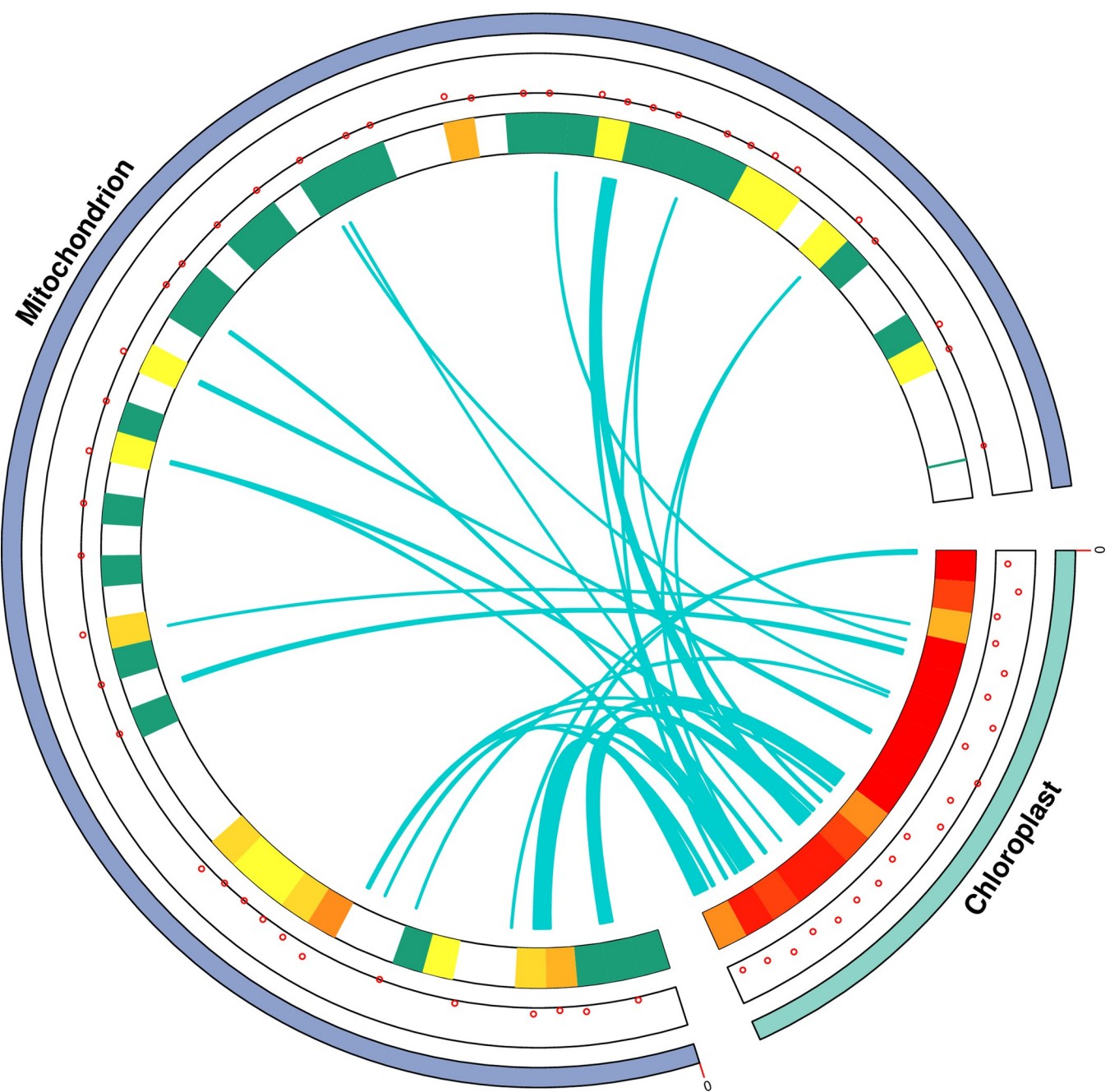

**Fig 5. Schematic representation of mitochondrial genome, chloroplast genome and chloroplast-like sequence of *M. integrifolia*.** Dots and heat maps inside the two chromosomes show where genes are located. The green lines in the circle show the regions of chloroplast-like sequences inserted from the chloroplast genome into the mt genome.

## 3.6 The substitution rates of protein-coding genes

In genetics, non-synonymous (Ka) and synonymous (Ks) substitution rates help understand the evolutionary dynamics of protein-coding genes among similar species since the Ka to Ks ratio indicates gene selection [72, 73]. In the present study, *N. nucifera* was used as a reference

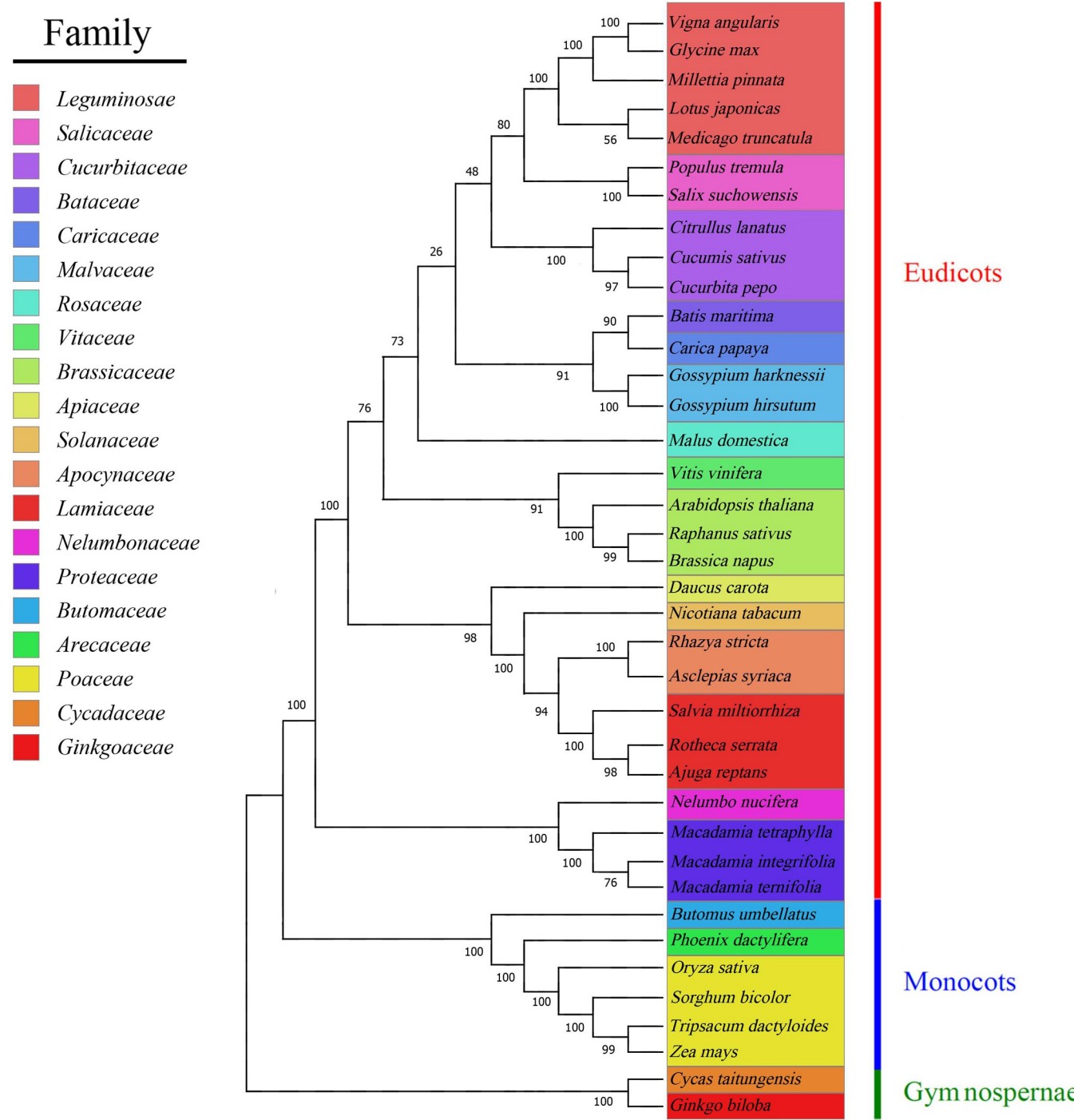

**Fig 6. The phylogenetic relationships of three species of *Macadamia* with other 35 plant species.** The Maximum Likelihood tree was constructed based on the sequences of 23 conserved protein-coding genes. Colors indicate the families that the specific species belongs.

species to calculate the Ka/Ks ratio of 40 protein-coding genes present in the mt genome of three *Macadamia* species. The Ks of *atp9* and *rps14*, and the Ka of *rps12* was 0. Besides, in most protein-coding genes, the Ka/Ks ratio was significantly less than 1 (Fig 7). However, the

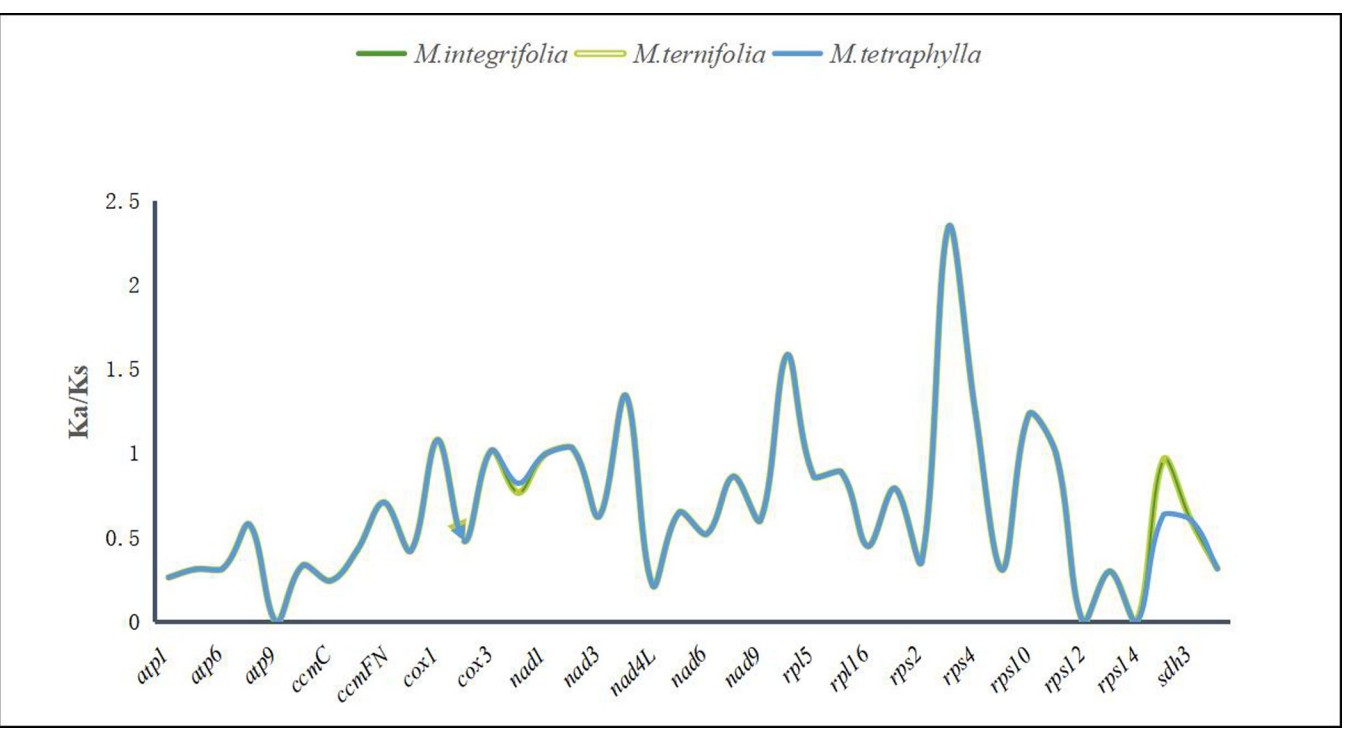

**Fig 7. The Ka/Ks values of 40 protein-coding genes of three *Macadamia* species.**

Ka/Ks ratio of *nad4*, *rpl2*, *rps3*, *rps4*, and *rps10* was greater than 1, with the *rps3* ratio being 2.34, implying that these genes might have undergone mutation related positive selection following *Macadamia* and *N. nucifera* differentiation from their last common ancestor [74]. Besides, the ATP synthase, Cytochrome C biogenesis, Ubiquinol Cytochrome C reductase, and Maturases of Ka/Ks ratios were below 1, implying that the negative selection acted on these genes (Table 2). Therefore, these genes may be highly conserved during the evolution of higher plants [75].

## 4. Conclusions

The complete mt genomes of *M. integrifolia*, *M. ternifolia* and *M. tetraphylla* share many common features with angiosperm mt genomes. In this study, we found that the mt genomes of the three *Macadamia* species were circular like most mt genomes. Compared them with the GC content of the mt genome of 36 other green plants, the results supported the conclusion that the GC content in the *Macadamia* species and angiosperms are highly conserved. In addition, we conducted studies on SSRs and longer tandem repeats in the three sets of data. Besides, 688 RNA editing sites were identified in 42 protein-coding genes, providing important clues for predicting gene function with new codons. By detecting gene migration, we observed 28 fragments (with five complete tRNA genes) were transferred from the cp genome to mt genome. The subsequent phylogenetic analysis results also showed their accuracy in plant classification. Moreover, based on the Ka/Ks substitution of protein-coding genes, most coding genes have undergone negative selection, indicating that the protein-coding genes in the mt genome are conserved in *Macadamia* species. The findings of this study provide information on the mt genome of *Macadamia* species, which is key in understanding the evolutionary history of the family Proteaceae.

## Supporting information

**S1 Table. The abbreviations and NCBI accession numbers of mt genomes used in this study.**
(XLSX)

**S2 Table. Perfect tandem repeats in the *Macadamia integrifolia* mitochondrial genome.**
(XLSX)

**S3 Table. Perfect tandem repeats in the *Macadamia ternifolia* mitochondrial gemone.**
(XLSX)

**S4 Table. Perfect tandem repeats in the *Macadamia tetraphylla* mitochondrial gemone.**
(XLSX)

## Author Contributions

**Conceptualization:** Shuo Wang, Chao Shi, Jin Liu.

**Data curation:** Yingfeng Niu, Yongjie Lu, Weicai Song, Xiyong He.

**Formal analysis:** Yingfeng Niu, Yongjie Lu, Weicai Song.

**Funding acquisition:** Ziyan Liu, Cheng Zheng, Shuo Wang.

**Investigation:** Yingfeng Niu, Yongjie Lu.

**Methodology:** Yingfeng Niu, Yongjie Lu, Xiyong He, Ziyan Liu.

**Project administration:** Xiyong He, Cheng Zheng, Chao Shi, Jin Liu.

**Resources:** Shuo Wang, Jin Liu.

**Software:** Yongjie Lu, Weicai Song, Xiyong He, Cheng Zheng, Shuo Wang.

**Supervision:** Ziyan Liu, Chao Shi, Jin Liu.

**Validation:** Yingfeng Niu, Xiyong He, Ziyan Liu, Cheng Zheng, Chao Shi.

**Visualization:** Weicai Song.

**Writing – original draft:** Yingfeng Niu, Yongjie Lu.

**Writing – review & editing:** Yingfeng Niu, Yongjie Lu.

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
