## [Decision Letter · Decision Letter 0]

14 Feb 2022

PONE-D-22-01656Assembly and comparative analysis of the complete mitochondrial genome of three Macadamia species (M. integrifolia, M. ternifolia and M. tetraphylla)PLOS ONE

Dear Dr. Shi,

Thank you for submitting your manuscript to PLOS ONE. After careful consideration, we feel that it has merit but does not fully meet PLOS ONE’s publication criteria as it currently stands. Therefore, we invite you to submit a revised version of the manuscript that addresses the points raised during the review process.

Please consider both reviewers comments carefully. Note the detailed comments of reviewer #1 are provided in the manuscript PDF file.

We look forward to receiving your revised manuscript.

Kind regards,

Yanbin Yin

Academic Editor

PLOS ONE

Journal Requirements:

"This work was supported by the National Natural Science Foundation of China (No.

31760215 and No. 31801022) and the Technology Innovation Talents Project of

Yunnan Province (2018HB086)."

"This work was supported by the National Natural Science Foundation of China (No. 31760215 and No. 31801022) and the Technology Innovation Talents Project of Yunnan Province (2018HB086)."

Reviewers' comments:

Reviewer's Responses to Questions

**Comments to the Author**

1. Is the manuscript technically sound, and do the data support the conclusions?

Reviewer #1: Yes

Reviewer #2: Yes

2. Has the statistical analysis been performed appropriately and rigorously? 

Reviewer #1: Yes

Reviewer #2: Yes

3. Have the authors made all data underlying the findings in their manuscript fully available?

Reviewer #1: Yes

Reviewer #2: Yes

4. Is the manuscript presented in an intelligible fashion and written in standard English?

Reviewer #1: Yes

Reviewer #2: Yes

5. Review Comments to the Author

Reviewer #1: Overall, the manuscript is well written and data/findings are clearly presented. The methods can be expanded and I have made notes on the manuscript to direct the authors on where to enhance the methods. Likewise, I have provided suggestions for additional literature that can be included in the manuscript at the discretion of the authors.

Reviewer #2: The authors presented a study on three mitogenomes from Macadamia species, and did the comparison of genomic features, repeat sequences, RNA editing, transfer from plastome into mitogenome and phylogeny with other higher plants. This manuscript provides some interesting findings. But there are still a few questions that need to be clarified and improved. In order to make the manuscript much clearer and the conclusions more valid, comments as follows:

1. The Figure 1 on the circular map of three Macadamia mitogenomes, the authors should remake these maps, clear labels for each gene.

2. I suggest the authors take care of the space and consistence on the number and word, for example, “31841bp (cox2)”, there is no space between “31841” and “bp”, but in other places there is space between them; also for the number, “31841” using “31,841” style, etc.

3. The other big problem is the references, the author must be very careful on every reference, such as uppercase, lowercase, italic, journal names, etc. The author must follow the instructions of the reference of this journal.

4. What are the structural differences among the three mitogenomes? How the author verify the accuracy of the three assemblies, especially for a couple of bps difference among these genomes?

5. In the part of “Up to 34.3% (236 sites) RNA editing sites occurred in the first base position of the codon, 68.6% (472 sites) appeared in the second base position, and there was no RNA editing in the third base position.”, why the total percentage > 100%?

6. For the Phylogenetic analysis, the authors need to provide more interesting information or findings.

7. I suggest the authors add more findings from the structural and evolution innovations from these mitogenomes, such as group I and II introns, etc.

6. PLOS authors have the option to publish the peer review history of their article (what does this mean?). If published, this will include your full peer review and any attached files.

Reviewer #1: No

Reviewer #2: No

---

## [Author Response · Author response to Decision Letter 0]

23 Mar 2022

Dear Reviewers:

Reviewer #1: Overall, the manuscript is well written and data/findings are clearly presented. The methods can be expanded and I have made notes on the manuscript to direct the authors on where to enhance the methods. Likewise, I have provided suggestions for additional literature that can be included in the manuscript at the discretion of the authors.

Thank you very much for your guidance. We have made necessary modifications according to the marks.

Reviewer #2: The authors presented a study on three mitogenomes from Macadamia species, and did the comparison of genomic features, repeat sequences, RNA editing, transfer from plastome into mitogenome and phylogeny with other higher plants. This manuscript provides some interesting findings. But there are still a few questions that need to be clarified and improved. In order to make the manuscript much clearer and the conclusions more valid, comments as follows:

1. The Figure 1 on the circular map of three Macadamia mitogenomes, the authors should remake these maps, clear labels for each gene.

The Figure 1 had been redrawn and uploaded in the new manuscript.

2. I suggest the authors take care of the space and consistence on the number and word, for example, “31841bp (cox2)”, there is no space between “31841” and “bp”, but in other places there is space between them; also for the number, “31841” using “31,841” style, etc.

We checked the full text and corrected similar errors.

3. The other big problem is the references, the author must be very careful on every reference, such as uppercase, lowercase, italic, journal names, etc. The author must follow the instructions of the reference of this journal.

All references have been corrected to Plos ONE style.

4. What are the structural differences among the three mitogenomes? How the author verify the accuracy of the three assemblies, especially for a couple of bps difference among these genomes?

As mentioned in the material method, depth of coverage was used to correct mitochondrial sequence information. SPAdes v.3.5.0 software was used to splice and assemble mt genome sequences. To correct the splicing results, the raw sequencing data were mapped to mitochondrial sequences using Geneious software. 

5. In the part of “Up to 34.3% (236 sites) RNA editing sites occurred in the first base position of the codon, 68.6% (472 sites) appeared in the second base position, and there was no RNA editing in the third base position.”, why the total percentage > 100%?

This statement has been modified in the new manuscript. 236 RNA editing sites occurred in the first base position of the codon, 472 sites appeared in the second base position, and there was no RNA editing in the third base position.

6. For the Phylogenetic analysis, the authors need to provide more interesting information or findings.

7. I suggest the authors add more findings from the structural and evolution innovations from these mitogenomes, such as group I and II introns, etc.

We have revised the above two points in the new manuscript. Thank you very much for your guidance

---

## [Decision Letter · Decision Letter 1]

20 Apr 2022

Assembly and comparative analysis of the complete mitochondrial genome of three Macadamia species (M. integrifolia, M. ternifolia and M. tetraphylla)

PONE-D-22-01656R1

Dear Dr. Shi,

We’re pleased to inform you that your manuscript has been judged scientifically suitable for publication and will be formally accepted for publication once it meets all outstanding technical requirements. Please fix the minor issues that the reviewer identified.

Kind regards,

Yanbin Yin

Academic Editor

PLOS ONE

Additional Editor Comments (optional):

Reviewers' comments:

Reviewer's Responses to Questions

**Comments to the Author**

1. If the authors have adequately addressed your comments raised in a previous round of review and you feel that this manuscript is now acceptable for publication, you may indicate that here to bypass the “Comments to the Author” section, enter your conflict of interest statement in the “Confidential to Editor” section, and submit your "Accept" recommendation.

Reviewer #1: (No Response)

Reviewer #2: All comments have been addressed

2. Is the manuscript technically sound, and do the data support the conclusions?

Reviewer #1: Yes

Reviewer #2: Yes

3. Has the statistical analysis been performed appropriately and rigorously? 

Reviewer #1: Yes

Reviewer #2: Yes

4. Have the authors made all data underlying the findings in their manuscript fully available?

Reviewer #1: Yes

Reviewer #2: Yes

5. Is the manuscript presented in an intelligible fashion and written in standard English?

Reviewer #1: Yes

Reviewer #2: Yes

6. Review Comments to the Author

Reviewer #1: I am pleased with the author's revision and attention to my comments. However, I have two very minor corrections for the authors to address:

Line 56: "Additionally,, they..."

Please remove the redundant comma.

Line 113 -114 : "...and NCBI were used to annotate the mt genome."

Please cite the program also; I apologize that my previous comment was vague regarding an NCBI citation. Did you use ORF finder? If so, please cite: “…and ORF finder (NCBI) were used to annotate the mt genome.”

Reviewer #2: There are minor errors in the manuscript:

1. Line 56, two comma after "Additionally";

2. Line 123, "Macadamia" is not italic.

7. PLOS authors have the option to publish the peer review history of their article (what does this mean?). If published, this will include your full peer review and any attached files.

Reviewer #1: No

Reviewer #2: No

---

## [Editor Report · Acceptance letter]

25 Apr 2022

PONE-D-22-01656R1 

Assembly and comparative analysis of the complete mitochondrial genome of three *Macadamia* species (*M. integrifolia*, *M. ternifolia* and *M. tetraphylla*) 

Dear Dr. Shi:

I'm pleased to inform you that your manuscript has been deemed suitable for publication in PLOS ONE. Congratulations! Your manuscript is now with our production department. 

Kind regards, 

on behalf of

Dr. Yanbin Yin 

Academic Editor

PLOS ONE